# Comparative Study on Wood CNC Routing Methods for Transposing a Traditional Motif from Romanian Textile Heritage into Furniture Decoration

Antonela Lungu [ID], Mihai Ispas, Luminiţa-Maria Brenci, Sergiu Răcăşan and Camelia Coşereanu *

Faculty of Furniture Design and Wood Engineering, Transilvania University of Brasov, B-dul Eroilor, nr. 29, 500036 Brasov, Romania; antonela.petrascu@unitbv.ro (A.L.); ispas_m@unitbv.ro (M.I.); brenlu@unitbv.ro (L.-M.B.); sergiu.racasan@unitbv.ro (S.R.)
\* Correspondence: cboieriu@unitbv.ro

**Abstract:** This paper presents experimental research on the Computer Numerical Control (CNC) routing of a traditional motif collected from Ţara Bârsei (Transylvania region) using two methods, namely, engraving (Engrave) and carving (V-Carve). The analysis of the CNC router processes includes the calculation of the path lengths, an assessment of the processing time and wood mass loss, and an evaluation of the tool wearing by investigating the tool cutting edge on a Stereo Microscope NIKON SMZ 18 before and after processing the ornament on wood. An aesthetic evaluation of the ornament routed on wood, using both the engraving and carving methods, is also conducted, whilst a microscopic analysis of the processed areas highlights the defects that occurred on the wood surface depending on the tool path.

**Keywords:** CNC router; textile heritage; stereo-microscopy investigation; traditional motif

## 1. Introduction

As in many other cultures in the world, the textile heritage from Romania is proof of cultural expression, and it has to be well preserved for the next generations. Unfortunately, the objects in this category may be subjected to degradation risks due to their advanced age and sometimes improper conservation [1]. Actual digital technologies supported by graphics programs, computer-aided design (CAD), and computer-aided manufacturing (CAM) allow for the replication and transfer of valuable motifs found on old textiles onto modern ones [2], thus contributing to their preservation. In light of the Framework Convention on the Value of Cultural Heritage for Society (Faro Convention), a "New heritage" is about to rise up, and it pertains to the use of the past in the present and its renewal into the future [3]. Taking advantage of modern technologies, creative industries, such as fashion, ceramics, the textile industry, the furniture industry, and others, are able to transpose old traditional motifs onto new objects, preserving them in new forms.

Computerized Numerical Control (CNC) routers in the furniture industry offer the possibility of milling decorations on the wood surface, importing the vector file made with graphic software, which renders the drawing of the original motif, or by converting 2D digital images into 3D representation through developed applications, such as Visual C++ software [4]. The sculpting process on a CNC router is based on optimization of the milling parameters, selection of the right tool, and the processing method. The majority of scientific papers investigate the quality of routed surfaces by measuring the roughness parameters [5–9] or by applying a mathematical model to adjust the feed rate depending on the surface quality along with the cutting direction and species of wood [10], or genetic algorithms on CAM software [11]. The spindle speed of 14,500 rpm and feed speed of 2 m/min for CNC routing birch and beech wood had as low effect values for the roughness parameter Ra (average surface roughness measured as the arithmetic averages of the absolute values for all deviations of peaks and ridges) for

a conventional way of cutting [5]. In the case of CNC milling of spruce and chestnut wood, 10,000 rpm for spindle speed and 5 m/min for feed speed were recommended for a minimum value of Rz (the average maximum peak to valley of five consecutive sampling lengths within the measuring length) roughness parameter, whilst for larch wood, a feed speed of 7 m/min and spindle speed of 18,000 rpm were found to be optimal [6]. Minimum Ra and Rz values were obtained for a spindle speed of approx. 17,000 rpm and a feed speed of 2 m/min in the case of CNC milling of Cedar of Lebanon pine [7].

Numerous studies were conducted to evaluate the tool wear or the effect of tool wear on the machining process for various species of wood or wooden-based materials [12–17]. Tool wear has a great influence on the quality of the product, its accuracy, and productivity. The measurement of the cutting edge profile gives indications about the tool wear level. It can be performed automatically [12] or by chemical, microscopic, and hardness analysis of the cutting knife. Indirect methods can also be used, such as the measurement of the electrical current consumption or the measurement of the sound pressure during processing. Both are higher when the tool wear increases [13]. Measuring Ra and Rz roughness parameters after successive CNC milling of pine wood, it was determined that, for the anti-wear coated tools, the values of the roughness parameters are higher in the phase of initial wear than those of an uncoated tool, but with increased wear, the situation is reserved and the roughness parameters for coated tools decreased to lower values than the initial ones [14]. Research on the correlation between the tool wear and cutting forces in the machining process of the Aleppo pine have shown that the edge recession occurred at a cutting length of 850 m, with abrupt wear at about 200 m [17].

The research presented in this paper shows the process of transposing a traditional motif taken from the textile heritage originating from the SE region of Transylvania from Romania (named Țara Bârsei) on the wood surface, as decoration such as for furniture, using a three-axis CNC router for the milling process. Two methods of transposing the motif were applied: engraving (Engrave) and carving (V-Carve), using V-Grooving Router Bit tool with an angled tip of 90°. Linux CNC version 2.7.0. software was used to simulate the machining process and the tool path for the two variants of the ornament. Maple wood was used for the experiment, and twenty successive ornaments were machined and executed for each variant of the model. The wear of the tools was determined by scanning electron microcopy of the tool cutting edge before starting the CNC milling process and after the ornaments were processed twenty times both by engraving and carving. A visual evaluation of the processed ornaments was conducted from the aesthetic and qualitative points of view, and details of the areas where wood fuzziness occurred were highlighted by stereo-microcopy analysis.

## 2. Materials and Methods

The traditional Romanian ornamentation is rich in geometric representations, characterized by a delicate refinement. The present research uses the plant motif from Figure 1 for the experiment, which is originally stitched in a cross technique on the sleeve of a traditional woman's blouse from the village of Bran, where Dracula's castle is located. The ornamental composition, subjected to the laws of symmetry and alternation, contains the motif of the clover leaf, the "water wave" pattern, and the fir leaf. All traditional motifs in Romania have a symbolic value. Thus, the clover suggests the magic number three, which exists in many ritual practices and generally protects human life, and the four-leaf clover is considered to bring luck [18]. Flowing water means a crossing, a test, and thus represents the passage, the flow of life, and the flow of time [19]. The fir tree can be assimilated with the tree of life, which is the backbone of the world. It is present at weddings and funeral rituals, and when it is represented with seven rows of branches, those are the seven heavens [20]. This popular motif with symbolic value was collected as a picture from the particular owner of the original blouse (Figure 1a) and then rendered in vector format using the professional vector graphics software CorelDRAW X17 (Figure 1b). The file was

then imported in AutoCAD LT 2017 (Figure 1c) and transferred to the 3-axis CNC router, ISEL GFV type, German production, with a maximum spindle speed of 15,000 rpm.

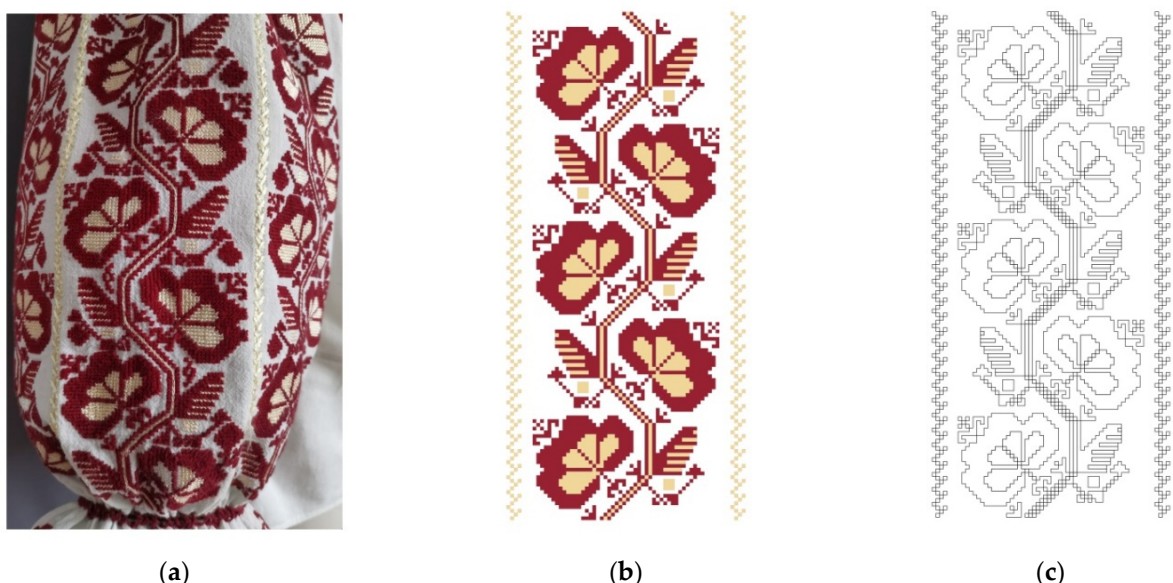

**Figure 1.** Traditional motif from Țara Bârsei subjected to the present research: (**a**) the picture of the textile object, (**b**) the digital vector format drawing, and (**c**) the contour of the ornament used by the CNC router for milling the maple wood surface.

The contour of the ornament (Figure 1c) was CNC machined in two ways: the first one was the engraving method (Engrave), for which a constant depth of the cut at 3 mm was applied to both closed contours and open contours of the drawing; the second one was the carving method (V-Carve), which can be applied only for closed contours, with variable depths between 1 mm to 3 mm for the surface and 3 mm for the contour. The other processing parameters of the CNC router were the spindle speed of 15,000 rpm and feed speed of 6 m/min.

Twenty maple (*Acer pseudoplatanus* L.) wood panels with dimensions of 300 mm × 200 mm × 11 mm, a moisture content of 11%, and a mean density of 615 kg/m$^3$ were used for the experiment. Due to its homogeneous structure and color compared to other species of wood, maple wood is recommended for carving.

The tool type chosen for the experiment is CMT Orange V-Grooving Router Bit angled 90° (Figure 2). These double cutting edge CMT bits offer a large range of woodworking possibilities, making clean cuts in the panels. They are made of super strength Fatigue Proof® steel with Tungsten carbide-tipped cutting edges [21].

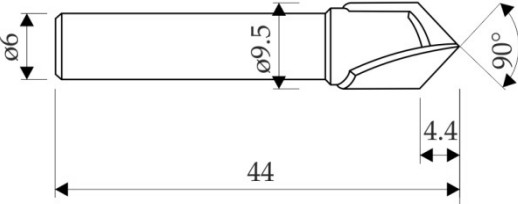

**Figure 2.** CMT Orange tool, V-Grooving Router Bit 90°, code 715.095.11.

### 2.1. Method to Analyse Tool Wear

In order to study the initial (abrupt) tool wear, twenty panels were used for CNC routing the ornament on the surface, and for each panel, the Engrave method was applied on one face and V-Carve on the other face. Two new V-Grooving 90° router bits were used for CNC milling of the wood surface: one for the Engrave method and the other for the V-Carve method. Before starting to process the ornaments on wood panels, the cutting

edge was studied using the stereo-microscope NIKON SMZ 18, with a zoom ratio of 18:1 and zooming range between 0.75× and 13.5×. The total magnification of the microscope used to investigate the tool cutting edge was 120×.

The stereo microcopy investigation was repeated after each new milling operation of the ornament on the wood surface for both CNC machining methods.

### 2.2. Simulation of Tool Path

The tool path was automatically calculated by the software Linux CNC version 2.7.0., a free software that runs under the Linux operating system. It was used together with the 3-axis CNC router. The software offers the possibility to simulate the machining process and to visualize the tool path and the data related to it: the length of the trajectory and the estimated processing time.

### 2.3. Method to Analyze the Wood Ornaments Produced

An investigation of the wood ornaments was conducted by comparing the two variants obtained by the Engrave and V-Carve methods from two points of view: the aesthetic one, following similarity with the original motif, and the quality of the processed surface analyzed by stereo-microscopy in the areas where wood fuzziness occurred. Wood mass loss after CNC machining the ornaments with the Engrave and V-Carve methods was also determined by weighing each panel before and after milling the wood. Wood mass loss may be correlated with the processing time in terms of productivity.

### 3. Results

The results obtained in the experiment are presented in Table 1.

**Table 1.** Recorded results for the Engrave and V-Carve methods.

| No. | Method | Mass of Removed Wood [1], in kg/Panel | Tool Path, in m/Panel | Processing Time, in min:s/Panel |
|---|---|---|---|---|
| 1 | Engrave | 0.038 (0.005) | 19.165 | 39:03 |
| 2 | V-Carve | 0.014 (0.002) | 49.741 | 59:48 |

[1] Average value calculated for the 20 panels on which experiment has been conducted. Values in the parenthesis are standard deviations.

### 3.1. Tool Wear

The wear occurred after processing the ornament twenty times on maple wood. This aspect was highlighted by the stereo-microscopy investigations. The tool cutting edge wear was not visible at the stereo-microscope after processing the ornament three times, but after milling the fourth panel, small changes were observed on the tool cutting edge (zone W, Figure 3; zone W rotated by 90°, Figure 4). Comparing the initial aspect of the tool cutting edge and the final one (after twenty uses), visible modifications occurred, both when the Engrave method (Figures 5 and 6) and the V-Carve method (Figures 7 and 8) were applied. A and B circled areas in Figures 5–8 were identified as the more affected zones of the cut edge, where its shape was modified as the result of the wear process.

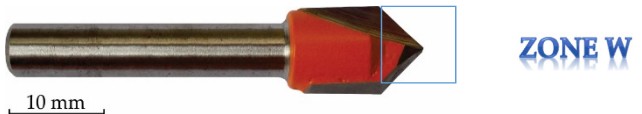

**Figure 3.** Zone W of the investigation on the wear of the tool cutting edge.

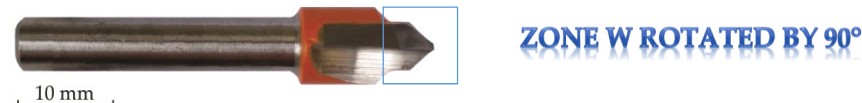

**Figure 4.** Zone W rotated by 90° for the investigation of the wear of the tool cutting edge.

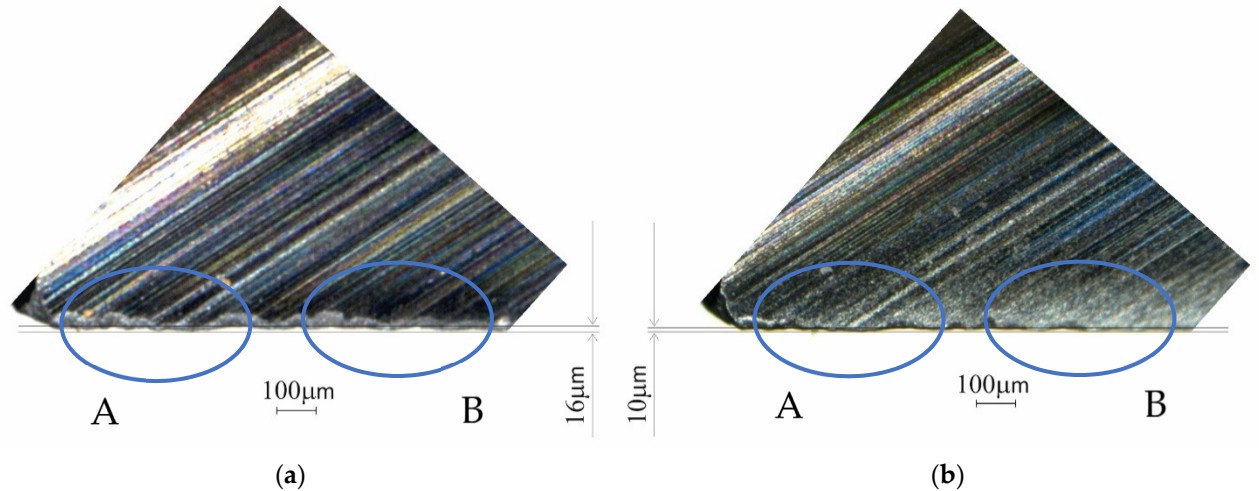

(**a**)　　　　　　　　　　　　　　　　　　　(**b**)

**Figure 5.** Zone W of the Engrave tool (120× magnification) on a stereomicroscope: (**a**) initial; (**b**) after twenty uses. A and B circled areas are the more affected one because of the wear process.

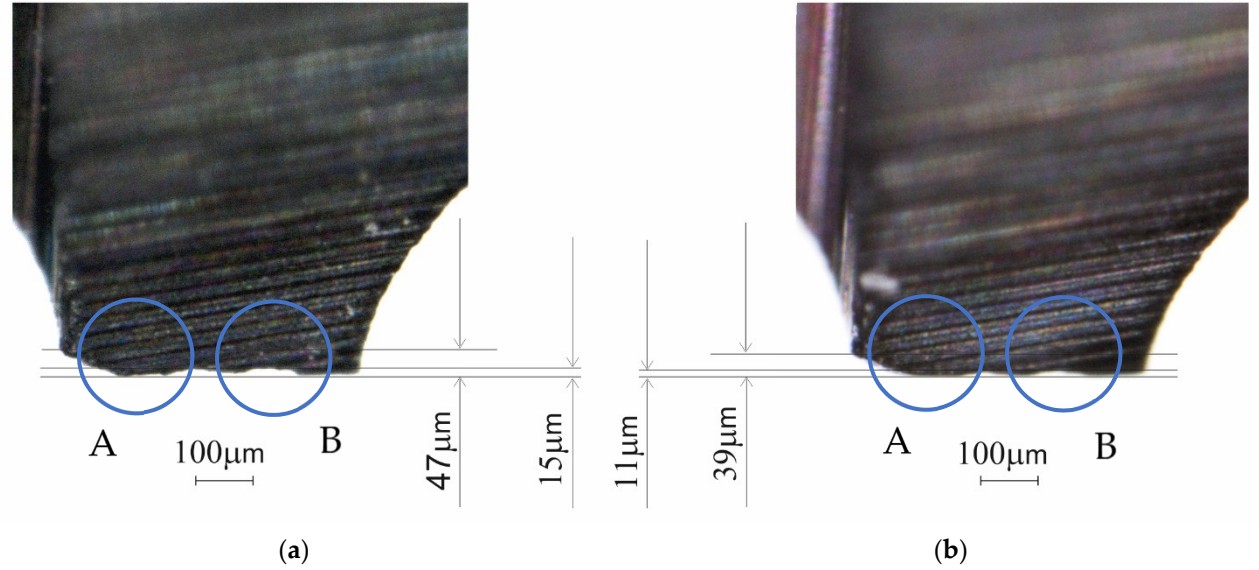

(**a**)　　　　　　　　　　　　　　　　　　　(**b**)

**Figure 6.** Zone W rotated by 90° of the Engrave tool (120× magnification) on a stereo-microscope: (**a**) initial; (**b**) after twenty uses. A and B circled areas are the more affected one because of the wear process.

The circles in Figures 5–8 indicate the zones where the differences are more visible (A and B circled areas). After twenty uses, the tools used for engraving and carving have blunt and rounded edges compared with their initial states in zone W and zone W rotated by 90°, especially for the tool tip area.

The wear process looks more pronounced for the second case, where the tool cutting edge was used for carving (V-Carve), as seen in Figures 7 and 8. Even if the tools were new, irregular contour of the cutting edge was noticed at the microscopic level before starting to use the tools for CNC routing the maple wood panels. Instead, the traces of sharpening the cutting edge were visible at this stage for zone W (Figures 5 and 7). The contour of the cutting edge became more regular after twenty uses of the tools, especially

when the V-Carve method was applied (Figures 7 and 8), but the traces of sharpened knife disappeared, replaced by a blunt-knife appearance.

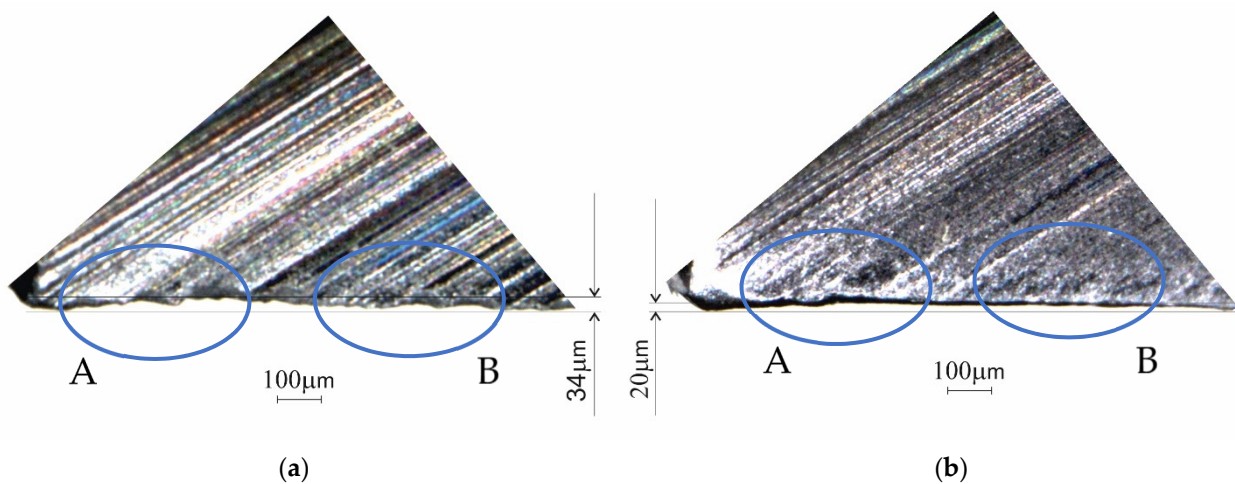

(**a**)　　　　　　　　　　　　　　　　　　(**b**)

**Figure 7.** Zone W of the V-Carve tool (120× magnification) on a stereo microscope: (**a**) initial; (**b**) after twenty uses. A and B circled areas are the more affected one because of the wear process.

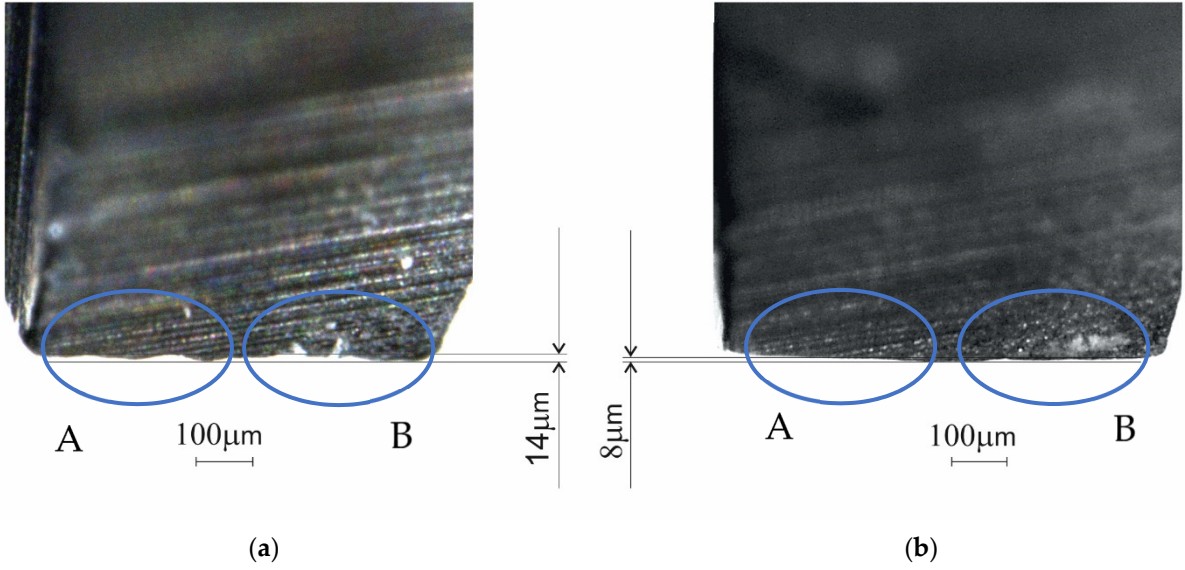

(**a**)　　　　　　　　　　　　　　　　　　(**b**)

**Figure 8.** Zone W rotated by 90° of the V-Carve tool (120× magnification) on a stereo-microscope: (**a**) initial; (**b**) after twenty uses. A and B circled areas are the more affected one because of the wear process.

### 3.2. Tool Path

In Figures 9 and 10, the tool path simulated by Linux CNC version 2.7.0. software is represented. The dashed line represents the trajectory traveled when idling, and a continuous line represents the path traveled during processing. With an interrupted red line, the shape and dimensions of the wooden piece are presented.

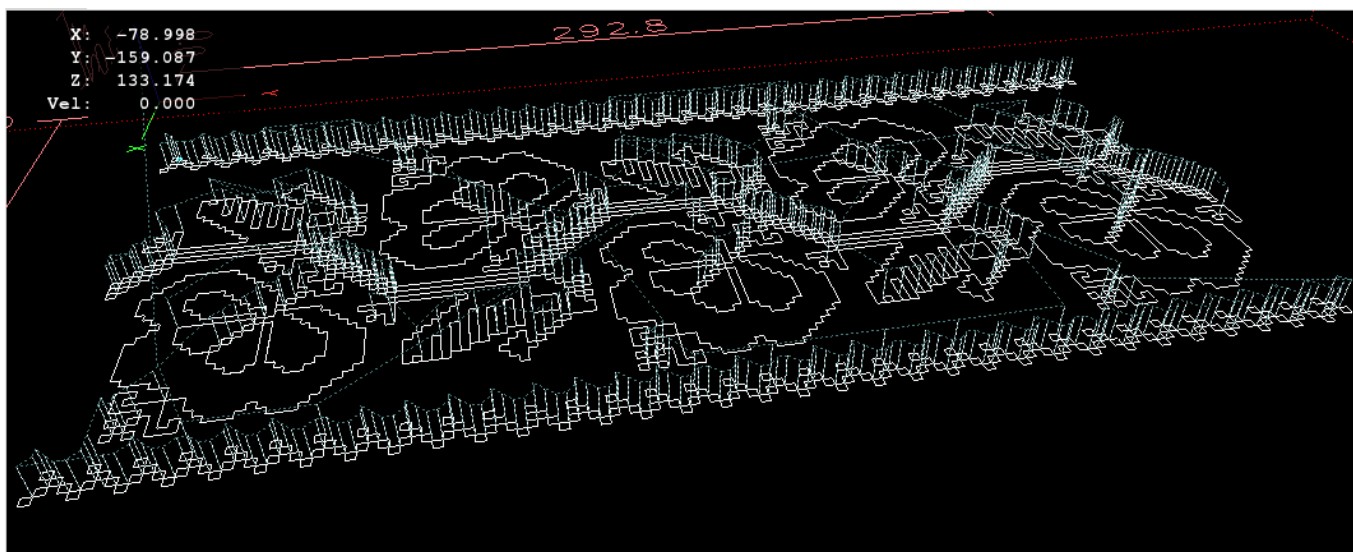

**Figure 9.** Tool path when milling with the Engrave method is applied.

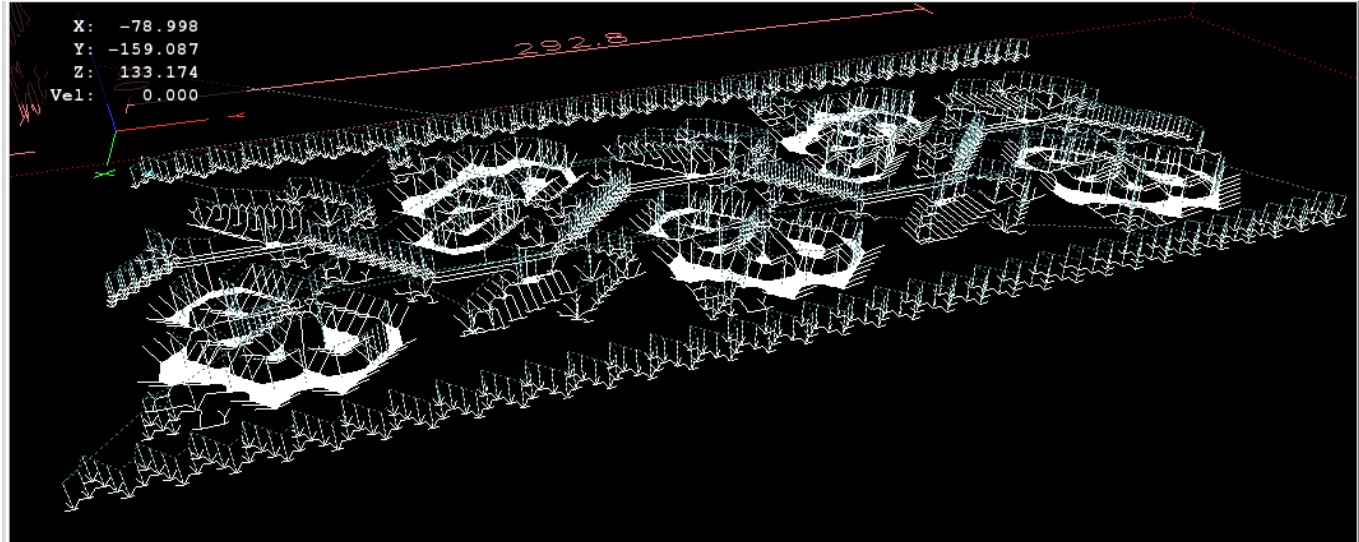

**Figure 10.** Tool path when milling with the V-Carve method is applied.

The calculated tool path by the Linux CNC software was 19,165.4 mm when the Engrave method was applied and 49,741.4 mm when the V-Carve method was applied for CNC routing. Calculating the whole length of the paths for the twenty processed ornaments, there was a route with 383.3 m length for engraving and 994.8 m length for carving. Similar research on Aleppo pine concluded that dge recession occurred at a cutting length of 850 m, with abrupt wear at about 200 m [17].

### 3.3. Wood Ornaments Investigation

A first evaluation of the ornaments was conducted by comparing the two variants transposed onto the wood surface with the original motif. The visual investigation revealed that the textile motif transferred by CNC milling on the maple wood surface looks different depending on the method applied: Engrave or V-Carve. In Figure 11a, the ornament looks crowded when applying the Engrave method. In the second case, when the V-Carve method was used (Figure 11b), the ornament processed on the wood surface resembled the original motif more.

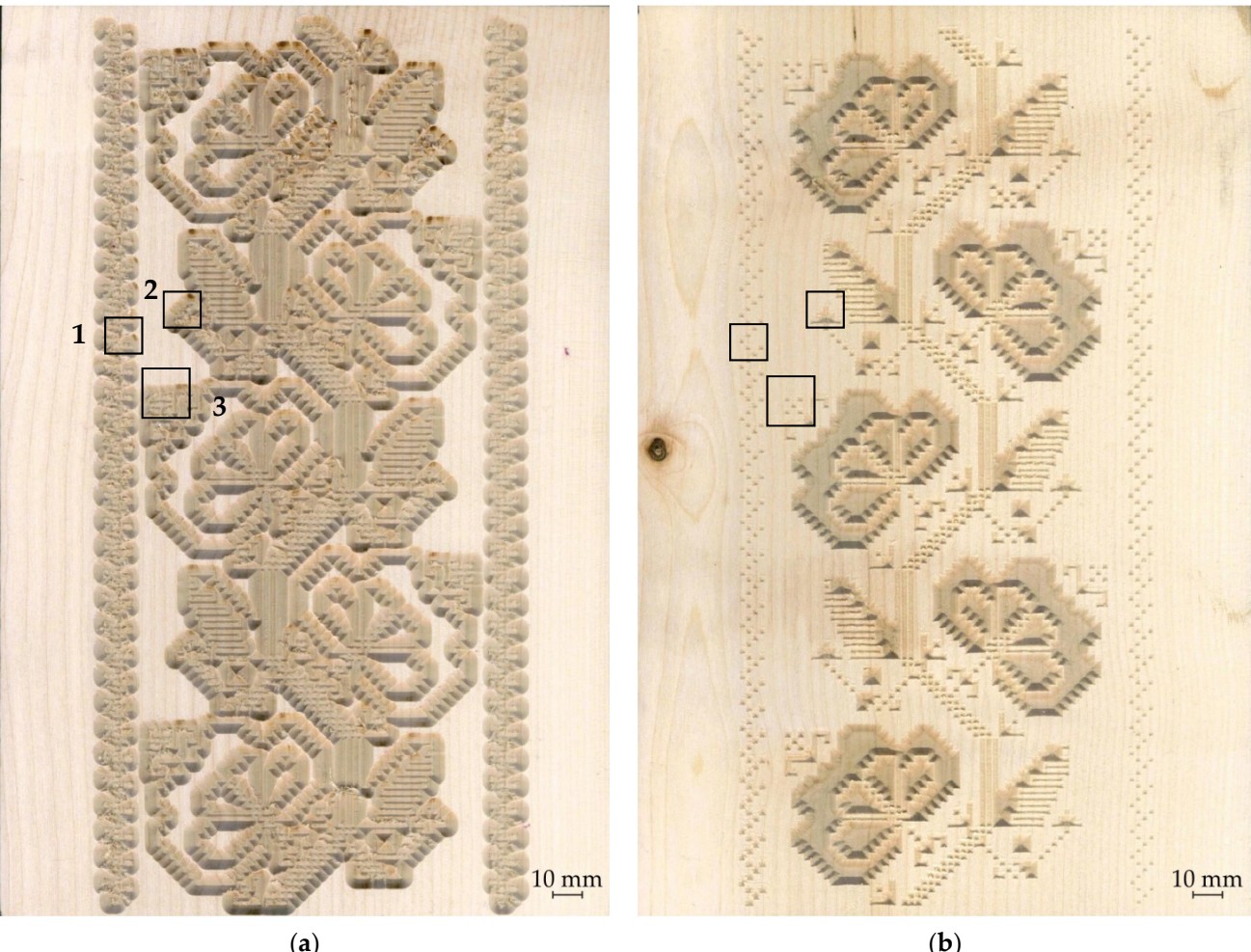

**Figure 11.** The CNC routed ornaments: (**a**) using the Engrave method; (**b**) using the V-Carve method. Areas marked with 1, 2 and 3 were identified to be characterized by pronounced wood fuzziness.

Studying the ornaments processed by the two methods, three zones where wood fuzziness occurred after processing the ornament by the Engrave method on the wood surface were highlighted. These zones are marked in Figure 11a, and they were assigned the numbers 1, 2, and 3. These zones were subjected to a stereo-microscopy analysis with 22.5× magnification and compared to similar zones of the V-Carved ornament. The images are presented in Figures 12–14 for the Engrave method (a) and for the V-Carve method (b). The arrows indicate fuzzy grains on the wood surface, more numerous and larger in size than on the variant of the ornament for which the Engrave method was used.

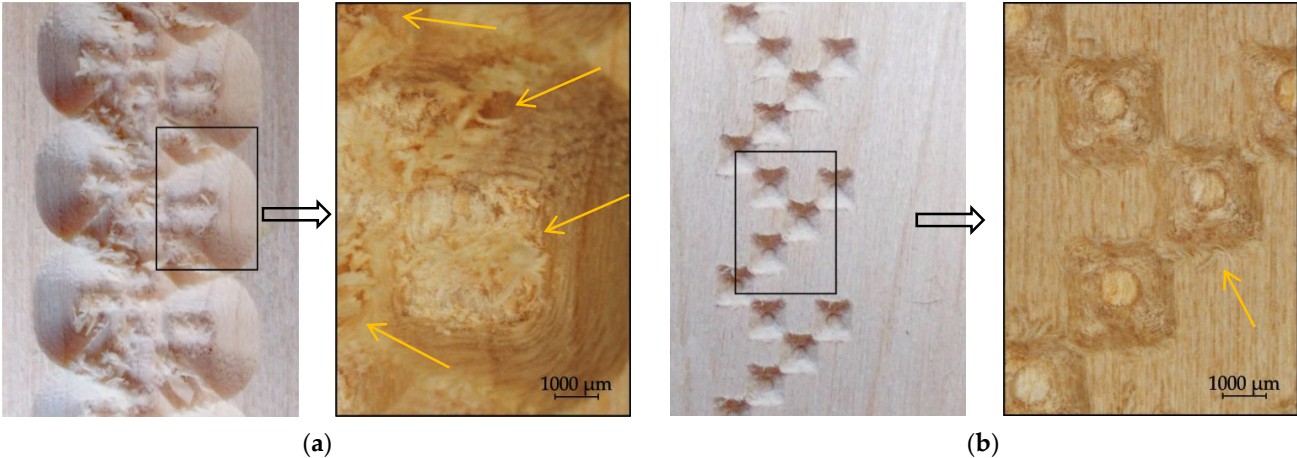

**Figure 12.** Detail 1 from Figure 8 subjected to stereo-microscopy analysis with 22.5× magnification: (**a**) ornament obtained by the Engrave method: detail from the picture (**left**) and the image with 22.5× magnification (**right**); (**b**) ornament obtained by the V-Carve method: detail from the picture (**left**) and the image with 22.5× magnification (**right**).

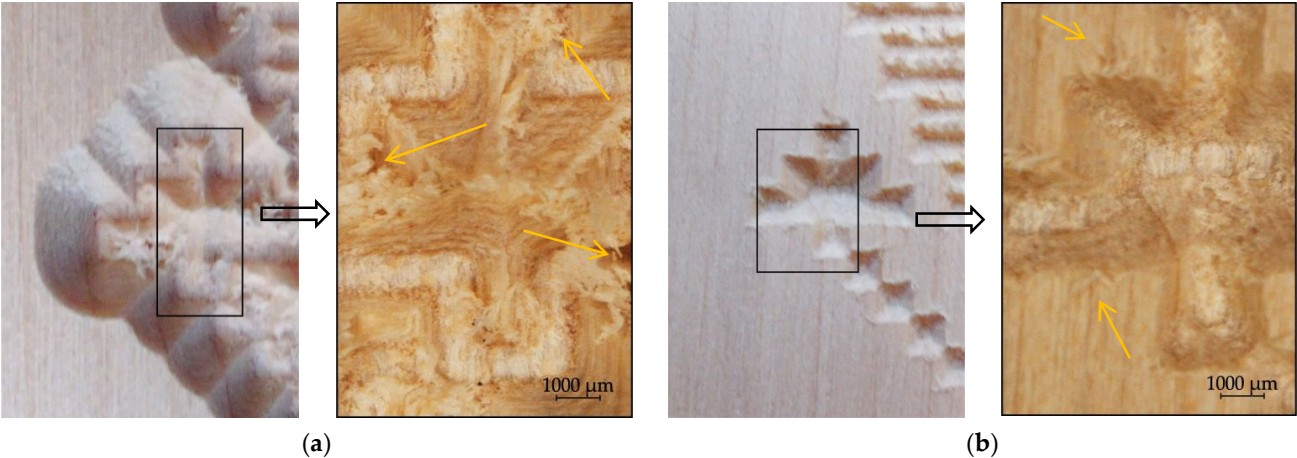

**Figure 13.** Detail 2 from Figure 8 subjected to stereo-microscopy analysis with 22.5× magnification: (**a**) ornament obtained by the Engrave method: detail from the picture (**left**) and the image with 22.5× magnification (**right**); (**b**) ornament obtained by the V-Carve method: detail from the picture (**left**) and the image with 22.5× magnification (**right**).

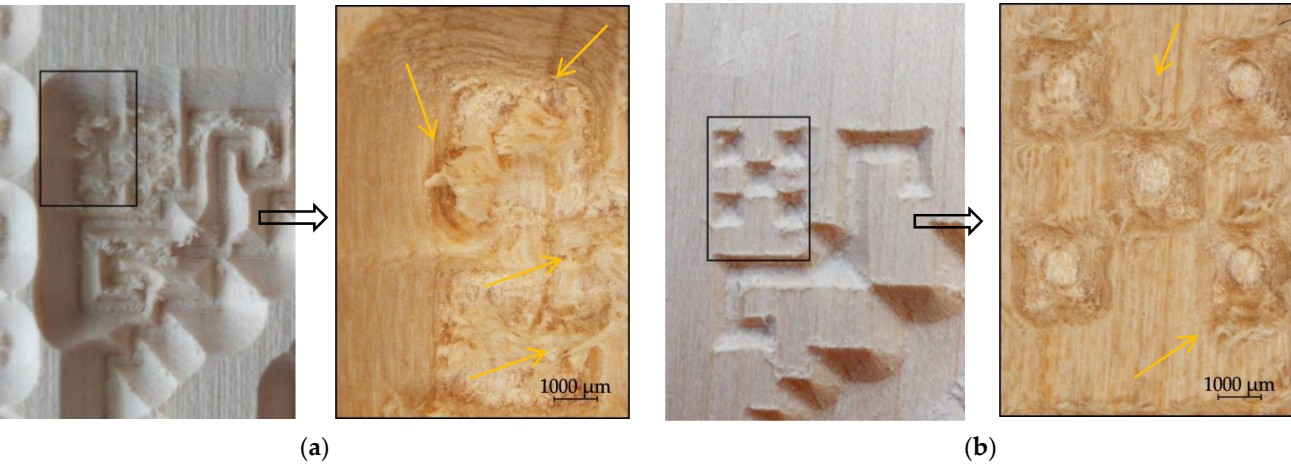

**Figure 14.** Detail 3 from Figure 8 subjected to stereo-microscopy analysis with 22.5× magnification: (**a**) ornament obtained by the Engrave method: detail from the picture (**left**) and the image with 22.5× magnification (**right**); (**b**) ornament obtained by the V-Carve method: detail from the picture (**left**) and the image with 22.5× magnification (**right**).

## 4. Discussion

For this research, a traditional Romanian motif taken from textile heritage was transposed as an ornament on maple wood panels by CNC machining with two different methods: engraving (Engrave) and carving (V-Carve) using the V-Grooving Router Bit angled at 90°. The comparison between the two processing methods included a visual analysis of the appearance of the processed ornament in the two variants, followed by a stereo-microscopy of the areas where fuzzy grains occurred after milling the wood; a tool wear assessment by microscopic investigation of the tool cutting edge after twenty successive millings of ornaments; and a calculation of the tool path and of the mass of the removed wood during the milling process and their correlation with the processing times.

The visual analysis revealed that the ornament looked totally different when it was engraved or carved by the CNC router. The carved ornament looked more similar the original motif. In addition, the microscopic investigation showed large and numerous wood fuzzy grains on the engraved surfaces, thus bringing another advantage to the V-Carve method.

The tool wear was highlighted by microscopic analysis of the tool cutting edge, where blunt and rounded edges were compared with their initial state after the ornament had been processed twenty times.

It is found that the wear of the tool used for engraving is 37.5% for the cutting area (see Figure 5) and 26.7% for the tip area (Figure 6). For the tool used for V-Carve milling, the wear is 41.2% for the cutting area (Figure 7), while for the tip, it was 42.8% (Figure 8). These data show that, in the case of milling with the Engrave method, the wear of the tool was less pronounced than that in the case of V-Carve milling. The explanation is related to the length of the tool path for the two applied methods (approx. 19.2 m when the Engrave method was applied and 49.7 m when the V-Carve method was applied).

It was also noticed that, when milling with the Engrave method, the wear affects the cutting edge area more, while when milling with the V-Carve method, the wear percentage of the tool tip is higher. This can be explained by the different contact areas between the tools and wood for the two cases, with the difference being the interior surface of the ornament, where the cutting depth varies between 1 mm to 3 mm, and the tip of the tool is involved in the milling process.

Correlations between tool path lengths calculated for twenty processed ornaments, their corresponding processing times, and masses of the removed wood are presented in Figure 15.

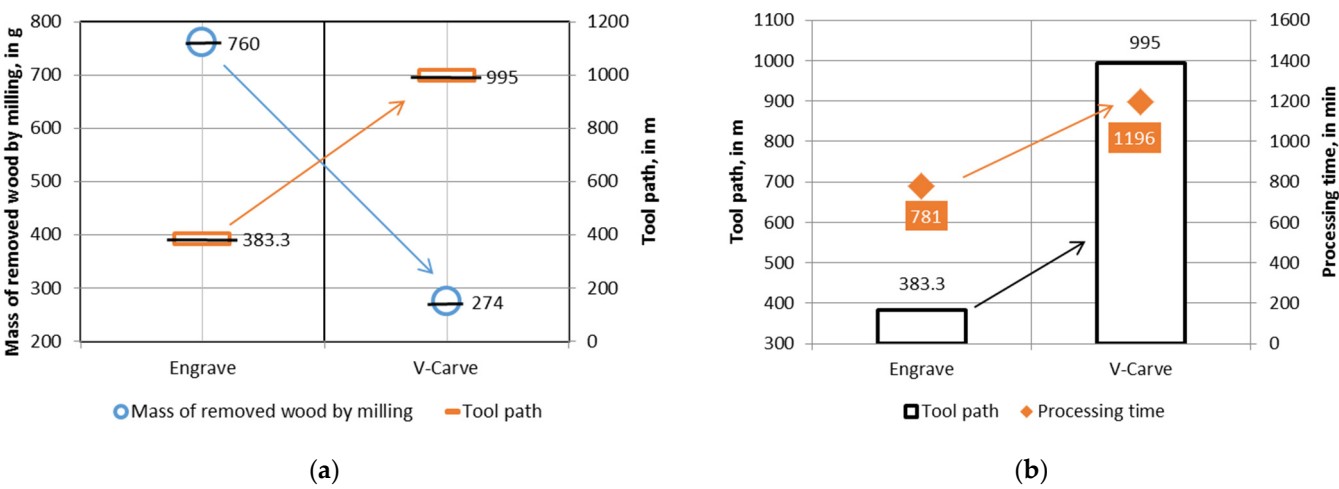

**Figure 15.** Comparison between the Engrave and V-Carve methods applied to CNC routing of an ornament on a maple wood surface: (**a**) correlation between the mass of removed wood and tool path length; (**b**) correlation between the tool path length and processing time.

## 5. Conclusions

The results of the present research show that the tool path is longer when applying the V-Carve method than for the Engrave method, and this aspect resulted in a longer processing time for the V-Carve method, as seen in Figure 15b. Instead, the mass of the wood removed during the routing process is higher when applying the Engrave method than when applying the V-Carve method. This can be explained by the constant cut depth of 3 mm for the Engrave method compared with the variable cut depth between 1 mm and 3 mm for the V-Carve method (3 mm only for the contour), resulting in a higher volume of wood removed in the first case. The longer tool path in the case of applying the V-Carve method, correlated with a longer processing time, is a drawback of this method in terms of productivity. Choosing the right method to be applied in CNC routing of ornaments similar to that used as an example in this research is a compromise between appearance, surface quality, and productivity. Being a furniture ornament, the aesthetic and the quality of the processed surface should be considered, so the V-Carve method is more advantageous from these points of view.

**Author Contributions:** Conceptualization, A.L., M.I. and C.C.; methodology, M.I. and C.C; software, A.L., L.-M.B. and S.R.; validation, A.L., M.I., L.-M.B., S.R. and C.C.; formal analysis, M.I.; investigation, A.L., L.-M.B. and C.C.; resources, A.L.; data curation, C.C.; writing—original draft preparation, A.L. and C.C.; supervision, C.C.; project administration, A.L.; funding acquisition, A.L. and C.C. All authors have read and agreed to the published version of the manuscript.

**Funding:** This research received no external funding.

**Institutional Review Board Statement:** Not applicable.

**Informed Consent Statement:** Not applicable.

**Data Availability Statement:** Not applicable.

**Acknowledgments:** The authors thank Dan Petrea for his work with milling processing, Melania Cristea for her help in measuring the weight of the panels and tools, and Gabriel Boriceanu for providing the photos with the model.

**Conflicts of Interest:** The authors declare no conflict of interest.

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
