# Peer review of "Comparative Study on Wood CNC Routing Methods for Transposing a Traditional Motif from Romanian Textile Heritage into Furniture Decoration"

_applsci, doi:10.3390/app11156713_

Round 1

Reviewer 1 Report

Proposals for changes and suggestions can be found in the attached file.

Author Response

ANSWERS TO REVIEWER 1

 Manuscript no.: 1287663

Article Title: Comparative Study on Wood CNC Routing Methods for Transposing a Traditional Motif from the Romanian Textile Heritage into Furniture Decoration

Authors: Antonela Lungu, Mihai Ispas, Luminiţa-Maria Brenci, Sergiu Răcăşan and Camelia Coşereanu

 Suggested Corrections:

Page nº 2 - Line 88

- After reference [18] is written "(p.661)" which should be removed.

Answer to Reviewer 1:

Thank you for your remarks. We have considered for the modified version of the article all your observations. The text was removed.

Page nº 3 - Line 117

- The numbering of subchapter "3.1." must be replaced by 2.1.

- The title “Tool wear” is the same title of page nº 5, line 149. Consider assigning a different title.

   (Suggestion: Method to analyse tool wear)

Answer to Reviewer 1:

The number and the title of subchapter was changed:

"3.1. Tool wear 2.1. Method to analyse tool wear”

Page nº 4 - Line 132

- The numbering of subchapter "3.2." must be replaced by 2.2.

- The title “Tool path” is the same title of page nº 6, line 168. Consider assigning a different title.

   (Suggestion: Simulation of tool path)

Answer to Reviewer 1:

The number and the title of subchapter was changed:

"3.2. Tool path 2.2. Simulation of tool path”

Page nº 4 - Line 138

- The numbering of subchapter "3.3." must be replaced by 2.3.

- The title “Wood ornaments investigation” is the same title of page nº 7, line 184. Consider assigning a different title.

   (Suggestion: Method to analyse wood ornaments produced)

Answer to Reviewer 1:

The number and the title of subchapter was changed:

"3.3. Wood ornaments investigation 2.3. Method to analyse wood ornaments produced”

Page nº 7 - Line 197

- Only the figure 9 of the page 8 is referred. It is necessary to mention the figures 10 e 11.

  (Suggestion: replace the text “The images are presented in figure. 9a for engrave method and in figure 9b for V-carve method” by “The images are presented in figure 9, 10 and 11, for engrave method (a) and for Vcarve method  (b)”)

Answer to Reviewer 1:

The text was changed:

"The images are presented in Figure 9 a for Engrave method and in Figure 9 b for V-Carve method. Figure 12, 13 and 14, for Engrave method (a) and for V-Carve method (b).”

2- General Comments

1) The meaning of the Ra and Rz roughness parameters is convenient to be included in the text (line 58, page 2).

Answer to Reviewer 1:

The meanings of Ra and Rz were included at lines 44 and 47:

"roughness parameter Ra (average surface roughness measured as the arithmetic averages of the absolute values of all the deviations of peaks and ridges) for a conventional way of cutting [5]. In the case of CNC milling of spruce and chestnut wood, 10000 rpm for spindle speed and 5 m/min for feed speed were recommended for a minimum value of Rz (the average maximum peak to valley of five consecutive sampling lengths within the measuring length)."

2) A scale included in the figures is preferable to photo magnification. Consider apply a scale to the figures 3; 4; 5; 8; 9; 10; 11.

Answer to Reviewer 1:

A scale was attached to each figure mentioned by the Reviewer 1.

 3) Tool weighing is not an effective method to assess tool wear in the number of tests performed.

Answer to Reviewer 1:

The Table 1 was completly changed:

Panel

no.

Phase

Panel weight,

in kg

Tool weight,

in kg ·10-3

Phase

Panel weight,

in kg

Tool weight,

in kg ·10-3

1

Before Engrave

0.406

12.082

Before V-Carve

0.363

12.083

After Engrave

0.363

12.082

After V-Carve

0.350

12.083

2

Before Engrave

0.395

12.082

Before V-Carve

0.350

12.083

After Engrave

0.350

12.082

After V-Carve

0.338

12.083

3

Before Engrave

0.407

12.082

Before V-Carve

0.373

12.083

After Engrave

0.373

12.082

After V-Carve

0.357

12.083

4

Before Engrave

0.412

12.082

Before V-Carve

0.372

12.083

After Engrave

0.372

12.081

After V-Carve

0.361

12.082

5

Before Engrave

0.412

12.081

Before V-Carve

0.377

12.082

After Engrave

0.377

12.081

After V-Carve

0.363

12.082

6

Before Engrave

0.399

12.081

Before V-Carve

0.361

12.082

After Engrave

0.361

12.081

After V-Carve

0.347

12.082

7

Before Engrave

0.406

12.081

Before V-Carve

0.373

12.082

After Engrave

0.373

12.081

After V-Carve

0.357

12.082

by the next table:

Table 1. Recorded results for Engrave and V-Carve methods.

No.

Method

Mass of removed wood1,

in kg/panel

Tool path, in m/panel

Processing time,

in min:s/panel

1

Engrave

0.038

19.165

39:03

(0.005)

2

V-Carve

0.014

49.741

59:48

(0.002)

    1 Average value calculated for 20 panels. Values in the parenthesis are standard deviations.

Also the text regarding the weighing of the tool were removed from lines 122-123, 129, 150-153, 216, 225-227:

"Before starting to process the ornaments on wood panels, the tools were weighed at an analytical balance with an accuracy of 0.001 g and the cutting edge was studied at the stereo-microscope NIKON SMZ 18, having a zoom ratio of 18:1 and zooming range between 0.75 x – 13.5 x. "

"Both the weighing of the tool and the The stereo microcopy investigation were repeated after each new milling operation of the ornament on the wood surface, for both CNC machining methods. "

"As seen in Table 1, the tool weight decreases with 0.001 g both for Engrave and V-Carve processing methods, which is too less to make a resolution on this data. But this small decrease show that the The wear occurred after processing the ornament twenty seven times on maple wood. This aspect was highlighted by the stereo-microscopy investigations. "

"The comparison between the two processing methods included a visual analysis of the appearance of the processed ornament in the two variants, followed by a stereo-microscopy of the areas where wood tearouts occurred after milling the wood; tool wear assessment after twenty seven successive milling of ornaments by weighing the tool and by microscopic investigation of the tool cutting edge; calculation of the tool path and of the mass of the removed wood during the milling process and their correlation with the processing time. "

"Only 0.001 g difference in tool weight resulted after milling the wood seven times to obtain the ornaments, no matter of the applied method. Even if this result didn’t focus attention on it, the The tool wear was highlighted by microscopic analysis of the tool cutting edge, where blunt and rounded edges compared to their initial state resulted after the ornament have been processed for twenty seven times. "

4) Contrary to what is stated on the line 157 of page 5, not visible modifications are observed in the figures 4a and 4b and in the figures 5a and 5b. 

Other analysis techniques or performing a larger number of tests would be useful to draw conclusions about the wear of the tool edge.

Answer to Reviewer 1:

A number of 13 additional experiments were carried out in order to draw a clear conclusion about the wear of the tool edge, so a total number of 20 experiments are now included in the final conclusion. Thus, the calculation made on line 180 was changed for the whole length, as follows:

 „Calculating the whole length of the paths for the twenty seven processed ornaments, a route of 134.2 383.3 m length resulted for engraving and 348.2 994.8 m for carving”.

In the same time, the Figures 4 and 5 (from lines 159 and 160) were changed with the final image of the tool edge after 13 more processing phases. Also, two more figures were added for the tool tip (when rotated with 90º), both for Engrave and V-Carve. As a result, the Figures caption received other numbers than before.

In the new microscopic images, more evident differences appear between the initial state of the tool edge and the final one.

5) I don´t agree with the following statement present in line 182 of page 7: “…it can be assumed that the experiment was conducted in the abrupt wear phase of the tools”, once there is no clear tool edge wear.

Answer to Reviewer 1:

The sentence was removed, and the differences are now more evident.:

"The calculated tool path by the Linux CNC software was of 19165.4 mm when Engrave method was applied and of 49741.4 mm when V-Carve method was applied for CNC routing. Calculating the whole length of the paths for the seven processed ornaments, a route of 134.2 m length resulted for engraving and 348.2 m for carving. Similar research on Aleppo pine concluded that the edge recession occurred at a cutting length of 850 m with the abrupt wear at about 200 m [17]. Based on this conclusion, it can be assumed that the experiment was conducted in the abrupt wear phase of the tools.”     

 Again, thank you for your remarks that improved the content and structure of the manuscript. Attached to these answers there is the original manuscript with highlighted corrections.   

The authors

Reviewer 2 Report

a) Tool wear should not be measured by weight change of the tool.

b) tool should be imaged new (before cutting), figures 4a and 5a indicated worn tools prior to tests.

c) the tools should also be photographed on both rake face and relief face.  Tool wear should be more obvious on the relief face.

d) need to explain the differences in tool path for engrave vs. V carve (how the tool move, depth of cut, etc.) in the approach section, not in the discussion section.  It is expected the a 1mm depth of cut will lead to better surface than  3mm, that is why V-carve is better than engrave

e) authors mentioned about Ra and Rz, as an important parameter, yet they did not measure the roughness.

f) there is no conclusion section.

Author Response

ANSWERS TO REVIEWER 2

Manuscript no.: 1287663

Article Title: Comparative Study on Wood CNC Routing Methods for Transposing a Traditional Motif from the Romanian Textile Heritage into Furniture Decoration

Authors: Antonela Lungu, Mihai Ispas, Luminiţa-Maria Brenci, Sergiu Răcăşan and Camelia Coşereanu

  1. a) Tool wear should not be measured by weight change of the tool.

Answer to Reviewer 2:

Thank you for your remarks. We have considered for the modified version of the article all your observations. The Table 1 was completly changed:

Panel

no.

Phase

Panel weight,

in kg

Tool weight,

in kg ·10-3

Phase

Panel weight,

in kg

Tool weight,

in kg ·10-3

1

Before Engrave

0.406

12.082

Before V-Carve

0.363

12.083

After Engrave

0.363

12.082

After V-Carve

0.350

12.083

2

Before Engrave

0.395

12.082

Before V-Carve

0.350

12.083

After Engrave

0.350

12.082

After V-Carve

0.338

12.083

3

Before Engrave

0.407

12.082

Before V-Carve

0.373

12.083

After Engrave

0.373

12.082

After V-Carve

0.357

12.083

4

Before Engrave

0.412

12.082

Before V-Carve

0.372

12.083

After Engrave

0.372

12.081

After V-Carve

0.361

12.082

5

Before Engrave

0.412

12.081

Before V-Carve

0.377

12.082

After Engrave

0.377

12.081

After V-Carve

0.363

12.082

6

Before Engrave

0.399

12.081

Before V-Carve

0.361

12.082

After Engrave

0.361

12.081

After V-Carve

0.347

12.082

7

Before Engrave

0.406

12.081

Before V-Carve

0.373

12.082

After Engrave

0.373

12.081

After V-Carve

0.357

12.082

The new table:

Table 1. Recorded results for Engrave and V-Carve methods.

No.

Method

Mass of removed wood1,

in kg/panel

Tool path, in m/panel

Processing time,

in min:s/panel

1

Engrave

0.038

19.165

39:03

(0.005)

2

V-Carve

0.014

49.741

59:48

(0.002)

    1 Average value calculated for 20 panels. Values in the parenthesis are standard deviations.

Also the text regarding the weighing of the tool were removed from lines 122-123, 129, 150-153, 216, 225-227:

"Before starting to process the ornaments on wood panels, the tools were weighed at an analytical balance with an accuracy of 0.001 g and the cutting edge was studied at the stereo-microscope NIKON SMZ 18, having a zoom ratio of 18:1 and zooming range between 0.75 x – 13.5 x."

"Both the weighing of the tool and the The stereo microcopy investigation were repeated after each new milling operation of the ornament on the wood surface, for both CNC machining methods."

"As seen in Table 1, the tool weight decreases with 0.001 g both for Engrave and V-Carve processing methods, which is too less to make a resolution on this data. But this small decrease show that the The wear occurred after processing the ornament twenty seven times on maple wood. This aspect was highlighted by the stereo-microscopy investigations."

"The comparison between the two processing methods included a visual analysis of the appearance of the processed ornament in the two variants, followed by a stereo-microscopy of the areas where wood tearouts occurred after milling the wood; tool wear assessment after twenty seven successive milling of ornaments by weighing the tool and by microscopic investigation of the tool cutting edge; calculation of the tool path and of the mass of the removed wood during the milling process and their correlation with the processing time."

"Only 0.001 g difference in tool weight resulted after milling the wood seven times to obtain the ornaments, no matter of the applied method. Even if this result didn’t focus attention on it, the The tool wear was highlighted by microscopic analysis of the tool cutting edge, where blunt and rounded edges compared to their initial state resulted after the ornament have been processed for twenty seven times."

A number of 13 additional experiments were carried out in order to draw a clear conclusion about the wear of the tool edge, so a total number of 20 experiments are now included in the final conclusion.

  1. b) tool should be imaged new (before cutting), figures 4a and 5a indicated worn tools prior to tests.

Answer to Reviewer 2:

Figures 4a and 5a indicate the image of the new tools. They were bought, unsealed and by microscope photographed. The magnification is 120 times.  Surprisingly, the new tools have unevenness that has blunted after processing.

  1. c) the tools should also be photographed on both rake face and relief face. Tool wear should be more obvious on the relief face.

Answer to Reviewer 2:

Two more figures were added for the tool tip (when rotated with 90º), both for Engrave and V-Carve. As a result, the Figures caption received other numbers than before.

In the new microscopic images, more evident differences appear between the initial state of the tool edge and the final one.

  1. d) need to explain the differences in tool path for engrave vs. V carve (how the tool move, depth of cut, etc.) in the approach section, not in the discussion section. It is expected the a 1mm depth of cut will lead to better surface than 3mm, that is why V-carve is better than engrave

Answer to Reviewer 2:

The differences between the two methods, Engrave and V-Carve  were explained in rows 99 to 104:

"The contour of the ornament (Figure 1 c) has been CNC machined in two ways: the first one was engraving method (Engrave), for which a constant depth of the cut of 3 mm was applied to both closed contours and open contours of the drawing; the second one was carving method (V-Carve), which can be applied only for closed contours, with variable depths between 1 mm to 3 mm for the surface and 3 mm for the contour. The other processing parameters of the CNC router were the spindle speed of 15000 rpm and feed speed of 6 m/min. "

  1. e) authors mentioned about Ra and Rz, as an important parameter, yet they did not measure the roughness.

Answer to Reviewer 2:

The measure of roughness is the next step of our research, as part of Antonela Lungu’s PhD thesis with the title:  Perspectives on the Development in Furniture of Some Traditional Romanian Motifs of Textile Heritage from Èšara Bârsei and Its Surroundings.

  1. f) there is no conclusion section.

Answer to Reviewer 2:

The title chapter Conclusion was inserted before line 235:

"5. Conclusions”

Moderate English changes required

Answer to Reviewer 2:

The paper was sent to an authorized translator and moderate English corrections were made.

Again, thank you for your remarks that improved the content and structure of the manuscript. 

The authors

Reviewer 3 Report

The paper presents experimental research on the CNC routing of a traditional motif collected from one particular Transylvania region (Å¢ara Bârsei) by the methods of engraving and carving. Indeed, there are not many published scientific papers dealing with heritage, preservation of traditional elements, CNC manufacturing technology, and design. That's why the value of this very concise and clear paper is very important, especially in the art and heritage preservation field. 

However, one correction is needed: there is no chapter Conclusion in the article. The title Conclusion is missing and should begin with line 235. Therefore I indicated that “it is not applicable” to the claim “whether the conclusion supports the results”. No additional corrections and reviews are required.

Author Response

ANSWERS TO REVIEWER 3

Manuscript no.: 1287663

Article Title: Comparative Study on Wood CNC Routing Methods for Transposing a Traditional Motif from the Romanian Textile Heritage into Furniture Decoration

Authors: Antonela Lungu, Mihai Ispas, Luminiţa-Maria Brenci, Sergiu Răcăşan and Camelia Coşereanu

The paper presents experimental research on the CNC routing of a traditional motif collected from one particular Transylvania region (Å¢ara Bârsei) by the methods of engraving and carving. Indeed, there are not many published scientific papers dealing with heritage, preservation of traditional elements, CNC manufacturing technology, and design. That's why the value of this very concise and clear paper is very important, especially in the art and heritage preservation field.

However, one correction is needed: there is no chapter Conclusion in the article. The title Conclusion is missing and should begin with line 235. Therefore I indicated that “it is not applicable” to the claim “whether the conclusion supports the results”. No additional corrections and reviews are required.

Answer to Reviewer 3:

Thank you for your remark. We have considered for the modified version of the article your observation. The title chapter Conclusion was inserted befor line 235:

"5. Conclusions”

English language and style are fine/minor spellcheck required

 Answer to Reviewer 3:

The paper was sent to an authorized translator and minor corrections were made.

Again, thank you for your remarks that improved the content and structure of the manuscript.

The authors

Round 2

Reviewer 2 Report

Tool wear between carve and engrave should be compared.  Authors did a decent job comparing weight lost in panel and processing time, what is still missing is the tool wear comparison.

Author Response

ANSWERS TO REVIEWER 2 (ROUND 2)

Manuscript no.: 1287663

Article Title: Comparative Study on Wood CNC Routing Methods for Transposing a Traditional Motif from the Romanian Textile Heritage into Furniture Decoration

Authors: Antonela Lungu, Mihai Ispas, Luminiţa-Maria Brenci, Sergiu Răcăşan and Camelia Coşereanu

  1. Tool wear between carve and engrave should be compared. Authors did a decent job comparing weight lost in panel and processing time, what is still missing is the tool wear comparison.

Answer to Reviewer 2:

Thank you for your remark. We have considered for the modified version of the article your observations.

The Figures 5, 6, 7 and 8 were replaced by new ones, where the measurements of the maximum uneveness heights were made, so to calculate the loss in height because of the tool wear as percentage of the initial height.  

The following explanation have been also introduced at the Discussion chapter, in order to have a comparison between the wear tool for milling with Engrave and V-Carve methods:

"It is found that the wear of the tool used for engraving is 37.5% for the cutting area (see Figure 5) and 26.7% for the tip area (Figure 6). For the tool used for V-Carve milling the wear is 41.2% for the cutting area (Figure 7) while for the tip it was 42.8% (Figure 8). These data show that in case of milling by the Engrave method, the wear of the tool was less pronounced than in case of V-Carve milling. The explanation is related to the length of the tool path for the two applied methods (approx. 19.2 m when the Engrave method was applied and 49.7 m when the V-Carve method was applied).

It was also noticed that when milling by the Engrave method the wear affects more the cutting edge area, while by the V-Carve method the wear percentage of the tool tip is higher. This can be explained by the different contact areas between the tools and wood for the two cases, the difference being for the interior surface of the ornament, where the cutting depth varies between 1 mm to 3 mm, and the tip of the tool is involved in the milling process."

Again, thank you for your remarks that improved the content and structure of the manuscript.   
